# Evaluating the Performance of Fire Rate of Spread Models in Northern-European *Calluna vulgaris* Heathlands

Charles D. Minsavage-Davis [1] and G. Matt Davies [2,*]

1   Department of Biology, Georgetown University, 37th and O Streets, N.W., Washington, DC 20057, USA; cd1231@georgetown.edu

2   School of Environment and Natural Resources, The Ohio State University, Kottman Hall, 2021 Coffey Road, Columbus, OH 43210, USA

*   Correspondence: davies.411@osu.edu

**Abstract:** Land-use, climate, and policy changes have impacted the fire regimes of many landscapes across northern Europe. Heathlands in oceanic climates are globally important ecosystems that have experienced an increase in the prevalence of destructive wildfire. Many of these landscapes are also managed using traditional prescribed burning that enhances their structural diversity and agricultural productivity. The changing role of wild and managed fire highlights a necessity to better understand the performance of fire behaviour prediction models for these ecosystems to support sustainable fire risk management. Our research evaluates the outputs of several empirical and quasi-empirical prediction models, as well as their varying software implementations, against observations of fire behaviour. The Rothermel model and its implementations predict rates of spread with similar accuracy to baseline empirical models and provide tolerable estimates of observed fire rate of spread. The generic shrubland empirical model developed by Anderson et al. consistently overpredicts observed rates of spread for prescribed burns in target fuel structures, but its predictions otherwise have a strong correlation with observed spread rate. A range of empirical models and software tools thus appear appropriate to assist managers who wish to evaluate potential fire behaviour and assess risk in heathland landscapes.

**Keywords:** fire behaviour; predictive modeling; Rothermel; empirical models; quasi-empirical models; United Kingdom; Europe; climate change; land-use change

## 1. Introduction

Fire is a common disturbance process in ecosystems, including boreal peatlands in Alaska, Canada, Fennoscandia, and Russia, as well as shrublands and forests in the British Isles, Scandinavia, and Mediterranean countries [1–8]. In many of these regions, climate change, fire suppression, and land-use change have, or are projected to increase, the incidence of wildfire and alter carbon emissions [8–10]. For example, climate change is expected to increase the number of wildfires across Italy [11]; Sweden has experienced an increase in wildfire activity as compared to records from before the industrial revolution [10]; in boreal forests, climate change impacts, such as severe droughts and vegetation change, are critical drivers of altered fire regimes [5,8,12,13]; and in Portugal, recent wildfires have impacted large areas of forest due to increasing extreme weather conditions [14]. Furthermore, fire suppression often has indirect social impacts such as in Fennoscandia where changes to boreal fire regimes have impacted traditional land usage by indigenous groups [15].

Uncontrolled wildfires can result in significant effects on economies, human health, and ecosystem function [16,17]. Because of this, ecologists increasingly acknowledge the fundamental role of fire in northern ecosystems (e.g., [18]) and seek to integrate this into management [19,20]. In oceanic northwestern Europe, shrublands, such as the heathlands and moorlands of the United Kingdom (UK) and Norway, hold globally important carbon

stores and have experienced a long history of both wild and managed fire [21–24]. Furthermore, traditionally managed burning has been a common tool in *Calluna vulgaris* L. Hull (heather, hereafter *Calluna*) dominated heathlands throughout Europe [24]. Such managed burning is argued to be important in promoting ecological processes, such as niche diversity [25–27] and managing fuels to reduce the impacts of severe wildfire [28]. Common burning practice involves lighting fires with comparatively short (~30 m) long fire fronts and burning strips that create a mosaic of fuel ages and habitat structures across landscapes. While we have limited data on the behaviour of wildfires in heathland fuels, experimental burns have demonstrated highly variable rates of spread and fireline intensities (1.3–12.6 m min$^{-1}$ and 137–4056 kW m$^{-1}$, respectively [21]). As a practice it has, however, diminished significantly in Norway [22] and is an increasingly contentious topic in the UK due to concerns about effects on water quality, ecosystem carbon stocks, and peatland carbon sequestration [5,29]. Nevertheless, there is concern over the potential for increased impacts from wildfires in northern heathlands due to a projected increase in severe fire weather [30], as well as changes in land-management policy (including restrictions on prescribed burning [31] and declines in traditional management). In Norway, unmanaged *Calluna*-dominated vegetation was found to experience rapid fire spread in the late winter and spring that has been exacerbated by climate change, dead fuel build-up [32], and drought [33]. To date, managers and policy makers in Europe have not been provided with adequate demonstration of the applicability of existing fire risk planning tools. This lack of knowledge and confidence limits the uptake of existing models and support systems which, in turn, leads to the potential to underestimate the changing fire risk environment [34].

Effective fire risk management requires a robust understanding of key controls on fire behaviour and the ability to produce reliable fire behaviour predictions. Thus, fire behaviour models have been developed for many different ecosystems across the world [35–37]. In the heathlands of northern Europe, fire ecology research has focused heavily on the life cycle, regeneration, and growth of *Calluna* [38–40]. Further important studies have quantified *Calluna* fuel chemistry [41], temperatures generated during *Calluna* fires [42], and fire rates of spread [43]. More recently, Davies et al. [44] recorded fire behaviour in experimental heathland fires in the UK, representing an array of fuel structures denoted by the growth phases of *Calluna* described by Gimingham et al. [45]. They were able to produce empirical models for rates of spread based on parameters, including the height of *Calluna*, fuel moisture, and wind speed. Subsequently, empirical models were produced for *Calluna* heathland fireline intensity and flame properties [21]. Away from heathland-specific analyses, generic empirical models of fire behaviour in shrub-dominated ecosystems have also been produced and cover a broad range of global climates [35]. These models aim to provide generalizable estimates of fire behaviour, but are not calibrated to ecosystem specific variations in fuel loadings, fuel composition and weather [35].

Empirically derived fire behaviour models can provide important guidance and knowledge for fire managers, yet are innately situation-specific and potentially misleading when extrapolated to new fuel, weather, or landscape conditions [46]. Thus, for example, the heathland fire behaviour models developed by Davies et al. are limited to flat terrain and a constrained range of weather conditions appropriate for prescribed burning [21,44]. In light of this alternative, more broadly applicable alternative models would be valuable. Fully physical-based approaches are often costly and require powerful computational hardware [47]. These methods, however, are becoming more viable as computational resources become more advanced and can provide intricate analyses of the effects of characteristics, such as fuel heterogeneity on fire behaviour [48]. In reality, there are trade-offs that come with using any fire behaviour modeling technique; the decision on which technique to use should be based on funding availability, time allotted for a given study, and the urgency of the situation for which fire is to be predicted, rather than on assumptions about the relative validity of any given model [49].

The Rothermel model [50] was a foundational development in the prediction of fire behaviour [36]. It provides a quasi-empirical approach combining relationships derived

from observed experimental fire behaviour data, as well as an appreciation of fundamental chemical and physical properties and processes relevant to fire behaviour [36,51]. In theory, the Rothermel model should be more broadly applicable than fuel-type specific empirical models, such as those produced by Anderson et al. [35] and Davies et al. [44], as a user is able to manipulate fuel chemistry, structure, and weather inputs. The Rothermel model has been widely applied and adopted, most frequently and extensively in the United States, to organize the deployment of firefighting resources and make fuel management decisions. A suite of software tools has been developed based on the Rothermel model that allows researchers and field managers to apply it more readily in their work [36]. BehavePlus provides instantaneous results for fire behaviour using constant conditions for fire weather but is typically less applicable for studies with larger spatial and temporal scales and variations [52]. FlamMap contains the tool FlamMap BASIC which expands BehavePlus predictions to a two-dimensional landscape to model spatial patterns of potential fire behaviour for constant weather and fuel moisture [53]. Farsite, also available within FlamMap, provides the ability to predict fire behaviour over time and across weather scenarios [54]. Finally, Vacchiano and Ascoli [55] translated the original Rothermel model into the package 'Rothermel' [56] for the statistical analysis software R [57]. Validation and testing are critical prior to the adoption of models in new systems to provide managers and policy makers with confidence in such decision support tools. Studies in a number or shrubland systems, including in the Mediterranean region of Europe and similar climates in California, have suggested that the Rothermel model and its associated software implementations can provide relatively robust predictions of shrubland fire behaviour [55,58–60]. To date, however, the Rothermel model and its derived tools have not been evaluated for oceanic heathland ecosystems in the UK and Norway. The objectives of this research were thus to: (i) evaluate the ability of key implementations of the Rothermel model to reproduce observed fire behaviour in *Calluna*-dominated heathland fuels; and (ii) to perform model sensitivity analyses to compare the behaviour of empirical models and Rothermel implementations across a representative range of fire-weather conditions for oceanic heathlands.

## 2. Materials and Methods

### 2.1. Comparing Modeled and Observed Fire Rate of Spread

Modeled fire rates of spread were compared with data collected during 27 experimental prescribed burns in *Calluna*-dominated heathland fuels (Table S1 in Supplementary Material). Full procedures for these experiments are described in Davies et al. [44]. Briefly, experimental fires (dimensions 15 m × 15 m or 20 m × 20 m) were burned in spring or late autumn on flat ground (<5% slope) in *Calluna* stand stages, representing the early-building, late-building, mature, and degenerate phases of the *Calluna* growth cycle [45]. Legg et al. qualitatively classified these plots into "High", "Medium", and "Low" fuel loading classes based on their *Calluna* stage and structural attributes [61]. Fuel structure, including fuel bed depth, the loading of live and dead fuels by size class and type (i.e., woody/herbaceous), and fuel bed bulk density, was monitored using destructive harvesting and the non-destructive FuelRule technique prior to the burns [62]. Fires were ignited using a driptorch to produce a line ignition along the upwind edge of the plots. *Calluna* comprised at least 90% of the overall fuel bed and the vast majority of the fuel was comprised of live shrubs. The *Calluna* canopy was generally underlain by a mat of pleurocarpous mosses and *Calluna* litter. These layers frequently had a very high fuel moisture content and experienced little consumption [63]. They were thus not considered as available fuel during our initial model testing. Fire weather conditions (wind speed at 1.5 m—mid-flame height, temperature, and humidity) were measured using a portable weather station located upwind of the burn. Samples of live and dead fuels were collected immediately prior to ignition to determine fuel moisture content. Data on dead fuel moisture content (FMC) for some fires was lacking; thus, we used the estimate for dead FMC provided by Legg et al. for these burns [61]. The fire rate of spread was measured by two methods: (i) a line of thermocouples located 5 m apart down the center of the plot; and

(ii) making visual estimates of the arrival time of the headfire at a series of measuring posts co-located with the thermocouples. The final assigned rate of spread was the mean of these two methods. The experiments captured a wide range of fuel, fire weather, and fire behaviour conditions, though they did not include the very dry conditions when wildfires are common. One-hour fuel loadings ranged from 0.90 t ha$^{-1}$ to 3.90 t ha$^{-1}$, live herbaceous fuel loadings from 4.00 t ha$^{-1}$ to 10.60 t ha$^{-1}$, live woody fuel loadings from 0.20 t ha$^{-1}$ to 8.90 t ha$^{-1}$, and fuel bed depths from 0.12 m to 0.51 m. Dead FMC ranged from 14.87 % to 29.00 %, live FMC from 55.19 % to 97.18 %, and wind speed from 5.83 km hr$^{-1}$ to 32.51 km hr$^{-1}$. Observed fire rates of spread ranged from 0.00 m min$^{-1}$ (i.e., no sustained spread) to 12.64 m min$^{-1}$. The known effect of fireline length on rate of spread [35] means that our estimated rates of spread may be conservative compared to the true quasi-steady-state spread rate for the given fuel structure. Our fires were, however, similar in size or larger than previous experimental shrub fires [64–67] and, critically, are of similar width to managed burns applied in *Calluna*-dominated heathlands [68].

We compared measured rates of spread with predictions from three different implementations of the Rothermel model: the 'ros' function of the R 'Rothermel' package (hereafter 'Roth') [56], BehavePlus [52], and Farsite [54]. Units for characteristics used as inputs for all Rothermel modeling tools were sourced from Andrews [36]. The Rothermel model and its implementations require a number of fixed chemical and physical fuel characteristics. Information for heathland fuels is limited but could be obtained from a variety of sources (Table 1). Using these constants, and the measured fuel characteristics associated with each individual experimental burn, we produced 27 individual fuel beds (Table S1 in Supplementary Material)—one for each burn. These were then used to generate the fire rate of spread predictions with each Rothermel implementation. Because Farsite produces temporally dynamic and spatially explicit predictions of fire behaviour, it was run on landscapes with flat ground and spatially and temporally constant inputs across 24 h. We used the maximum rate of spread as the estimate of steady-state head fire rate of spread.

**Table 1.** Input constants for Roth, BehavePlus, and Farsite. Live herbaceous fuels were combined, based on relative sizes, with 10 and 100 h fuels due to a lack of representation. SAV is surface-area-to-volume ratio. Table adapted from Legg et al. [61].

| Input Description | Value | Data Source |
|---|---|---|
| 1 Hour SAV (m$^{-1}$) | 9560 | EUFirelab [69] |
| 10, 100-Hour and Live Herbaceous SAV (m$^{-1}$) | 8810 | EUFirelab [69] |
| Live Woody SAV (m$^{-1}$) | 1000 | EUFirelab [69] |
| *Calluna* Heat of Combustion (kJ kg$^{-1}$) | 20,810 | Hobbs [41] |
| Moisture of Extinction (%) | 30 | Legg et al. [61] |

To provide a baseline against which to evaluate the performance of the Rothermel implementations, we examined rate of spread predictions from three empirical models (Equations (1)–(3)): rate of spread models 1 and 2 from Davies et al. (hereafter Davies 1 and Davies 2, Equations (1) and (2)) [44] and the generalized shrubland rate of spread model from Anderson et al. (hereafter Anderson, Equation (3)) [35].

$$\text{ROS} = 0.791 + 7.917\, h^2\, U \tag{1}$$

$$\text{ROS} = 8.304 + 7.286\, h^2\, U - 0.097\, M1 \tag{2}$$

where ROS is predicted steady-state rate of spread (m min$^{-1}$), $h$ is mean *Calluna* height (m), $U$ is 10–20 m wind speed upwind of fires (m s$^{-1}$), and $M_1$ is canopy fuel moisture content that translates to live woody moisture (%).

$$\text{ROS} = 6.421\, U_2{}^{0.9942}\, h^{0.3722} \exp(-0.0761\, M_d) \exp(-0.003131\, M_l) \tag{3}$$

where ROS is predicted steady-state rate of spread (m min$^{-1}$), $U_2$ is wind speed at 2 m height (km hr$^{-1}$), $h$ is mean vegetation height (m), $M_d$ is dead fine fuel moisture content (%), and $M_l$ is live fine fuel moisture content (%).

Some of these models (Equations (1) and (2) [44]) were developed with the same experimental fire data used to evaluate the Rothermel implementations. Our analysis thus does not provide any independent verification of these empirical models. Rather, these analyses provide context against which to consider the predictive ability of the other models. Anderson et al.'s model was developed based on shrubland fires with a minimum fire front length of 50 m and they demonstrated that smaller fires are unlikely to reach maximal quasi-steady-state rate of spread for the given fuel type [35]. When testing Equation (3), we therefore examined predictions compared to our original spread values and to the steady-state rate of spread estimated using the relationship between the ignition line length and the spread rate also presented in Anderson et al. [35] (Equation (4)).

$$R_{ss} = R \, (1 + 9 \exp(-0.00316 L^2)) \tag{4}$$

where $R_{ss}$ is quasi-steady-state rate of spread (m min$^{-1}$), R is measured rate of spread (m min$^{-1}$), and L is ignition line length (m)

All statistical analyses were performed in R [57]. To evaluate agreement between predicted and observed rates of spread, root mean squared error (RMSE) was calculated using the 'rmse' function of the 'Metrics' package in R [70]. Linear models were then generated to examine the strength and pattern of association between predicted and observed spread rates. Model predictions were assessed by testing for significant deviation between the slopes of the predicted vs. observed regression line and the 1:1 line of perfect agreement (LPA) [71]. We used Equation (5) to calculate t-values which were then compared to a table of critical values for n − 2 degrees of freedom. A potentially significant over- or under-prediction was indicated when the slope of the regression line differed significantly from the 1:1 line. The position of the 1:1 line relative to the 95% confidence intervals of the regression line was also examined. Where the majority of the 1:1 line fell within a model's 95% confidence intervals, it was deemed that predicted and observed results were not significantly different from each other.

$$t = (1 - b) \, SE_b^{-1}, \tag{5}$$

where $b$ is the coefficient for a regression line, $SE_b$ is the standard error of that regression, and t is a critical value score.

Finally, we used a two-way ANOVA ('aov' function in base R) to test for differences in predicted fire behaviour as a function of fuel class (High, Medium, Low), fire behaviour model and their interaction [57]. We were particularly interested in results for the latter which would indicate that predictions for a particular fuel class were contingent upon the fire behaviour model used. For any significant model terms, post-hoc pairwise comparisons were performed using the 'TukeyHSD' function in base R [57].

### 2.2. Comparing Fire Model Performance

Rate of spread predictions were made for varying fuel structures, fuel moistures. and wind speeds—variables known to be critical controls on rate of spread in *Calluna*-dominated fuels [44]. Spread rates were predicted for the High, Medium, and Low Rothermel fuel beds described by Legg et al. for *Calluna* heathlands (Table S1 in Supplementary Material) [61]. A representative range of live fuel moisture and wind speed conditions was developed based on data reported in Davies et al. [72]. Specifically, five steps of live woody FMC, ranging from 55 to 95% (10% increments), were combined with ten steps of wind speed, ranging from 5 to 23 km hr$^{-1}$ (2 km hr$^{-1}$ increments) to create a suite of fifty fire weather scenarios that broadly represent conditions encountered during the legal managed burning season in the UK [61]. Rate of spread was predicted for each combination of fuel bed, fuel moisture. and wind speed using all of the Rothermel implementations and empirical models.

To evaluate differences in model performance across the broader range of FMC and wind speed conditions we used (i) an ANOVA (interacting effects of fuel class and fire behaviour model; 'aov' function in base R) [57] to test for differences in the mean predicted spread rate between models and between models within fuel classes; (ii) a Levene's test, using the 'leveneTest' function of the 'car' package in R [73] to assess differences in spread rate variances between models within each fuel class; and iii) a sensitivity analysis to evaluate the independent effects of changes in fuel class, live FMC, and wind speed on rate of spread.

## 3. Results

Predicted fire rates of spread for all 27 fuel structures, collected alongside prescribed fires, varied both within and between the empirical models and Rothermel implementations (Table S2 in Supplementary Material). For the empirical models, predictions from Davies 1 ranged from 1.23 m min$^{-1}$ to 9.78 m min$^{-1}$ (mean $\pm$ 1 SD = 3.98 $\pm$ 2.53 m min$^{-1}$), Davies 2 from 0.11 m min$^{-1}$ to 11.23 m min$^{-1}$ (mean $\pm$ 1 SD = 3.99 $\pm$ 2.75 m min$^{-1}$), Anderson from 4.21 m min$^{-1}$ to 27.65 m min$^{-1}$ (mean $\pm$ 1 SD = 10.97 $\pm$ 6.82 m min$^{-1}$), Roth from 0.48 m min$^{-1}$ to 12.87 m min$^{-1}$ (mean $\pm$ 1 SD = 3.53 $\pm$ 3.19 m min$^{-1}$), BehavePlus from 0.06 m min$^{-1}$ to 9.99 m min$^{-1}$ (mean $\pm$ 1 SD = 2.81 $\pm$ 2.65 m min$^{-1}$), and Farsite from 0.05 m min$^{-1}$ to 8.25 m min$^{-1}$ (mean $\pm$ 1 SD = 2.37 $\pm$ 2.18 m min$^{-1}$) versus observed values from 0 m min$^{-1}$ to 12.64 m min$^{-1}$ (mean $\pm$ 1 SD = 3.98 $\pm$ 3.34 m min$^{-1}$).

### 3.1. Comparing Modeled and Observed Fire Rate of Spread

When compared to the observed rates of spread from the experimental fires, RMSE was, unsurprisingly, lowest for the Davies 1 and Davies 2 empirical models which were built using this same dataset (2.13 m min$^{-1}$ and 1.83 m min$^{-1}$, respectively). These results thus provide a baseline against which to judge the performance of the other models. While overall RMSE was substantially higher for the Anderson empirical model (8.64 m min$^{-1}$), those for Roth (2.64 m min$^{-1}$), BehavePlus (2.92 m min$^{-1}$), and Farsite (3.09 m min$^{-1}$) were fairly similar to the baseline expectation provided by the Davies models. When examining the performance of the Anderson model, correcting our measured rates of spread to the estimated steady-state spread rate led to an increase in RMSE of 8.39 m min$^{-1}$. When considering the three fuel structure classes separately, RMSEs were generally greatest within the "High" fuel load class while values were lower for the "Medium" and "Low" fuel classes for all fire behaviour models except Anderson.

For all fire behaviour prediction models, slopes of the regressions of predicted versus observed rates of spread did not significantly differ from the 1:1 line (Table 2 and Figure 1). Regression $R^2$ values ranged from 0.39 for BehavePlus and Farsite to 0.69 for the baseline provided by Davies 2. Anderson generated consistent and substantial over-predictions resulting in spread rates approximately twice those observed in the field. Anderson's performance was, however, otherwise strong with a slope matching the 1:1 and a relatively high $R^2$ (0.46), suggesting it provides comparatively precise, but biased, estimates. With steady-state correction of our observed rates of spread, the substantial overprediction by Anderson was reduced and the regression line shifted below the 1:1 (Figure S1 in Supplementary Material). However, $R^2$ for the resulting regression was much lower at 0.31. All other fire behaviour models tended to underpredict at higher rates of spread with the 1:1 line generally falling above the 95% confidence intervals of the regression line (Figure 1 and Table 2). This was most noticeable for the BehavePlus and Farsite implementations of the Rothermel model but was less extreme for the R version (Roth).

**Table 2.** Performance of each modeling technique versus observed fire behaviour. 'Model' is fire behaviour linear model. '$R^{2}$' is the adjusted correlation for each regression. '*b*' is the coefficient of each regression. 'df' is n − 2 degrees of freedom. '*t*' is the test statistic for performance testing. '$p_{slope}$' is a significance value for differences between the slope of each regression line and the 1:1 LPA, based on the t-statistic. For any non-significant *p*-value based on α = 0.05, 'LPA' represents where the LPA resides relative to the 95% confidence intervals of each regression line: W is within, A is above, and B is below. Corrected is Anderson versus corrected observations of rates of spread for ignition line lengths used in our experimental burns from Equation (4) [35].

| Linear Model | $R^2$ | *b* | Intercept ± Error | df (n − 2) | t | $p_{slope}$ | LPA |
|---|---|---|---|---|---|---|---|
| Davies 1 | 0.58 | 0.58 | 1.69 ± 0.51 | 25 | 0.25 | 0.80 | A |
| Davies 2 | 0.69 | 0.68 | 1.27 ± 0.48 | 25 | 0.20 | 0.84 | A |
| Anderson | 0.46 | 1.39 | 5.46 ± 1.55 | 25 | −0.08 | 0.94 | B |
| Roth | 0.45 | 0.60 | 0.94 ± 0.68 | 25 | 0.15 | 0.88 | A |
| BehavePlus | 0.39 | 0.50 | 0.69 ± 0.64 | 25 | 0.23 | 0.94 | A |
| Farsite | 0.39 | 0.41 | 0.64 ± 0.53 | 25 | 0.34 | 0.94 | A |
| Corrected | 0.31 | 0.27 | 7.72 ± 1.74 | 25 | 0.14 | 0.89 | A |

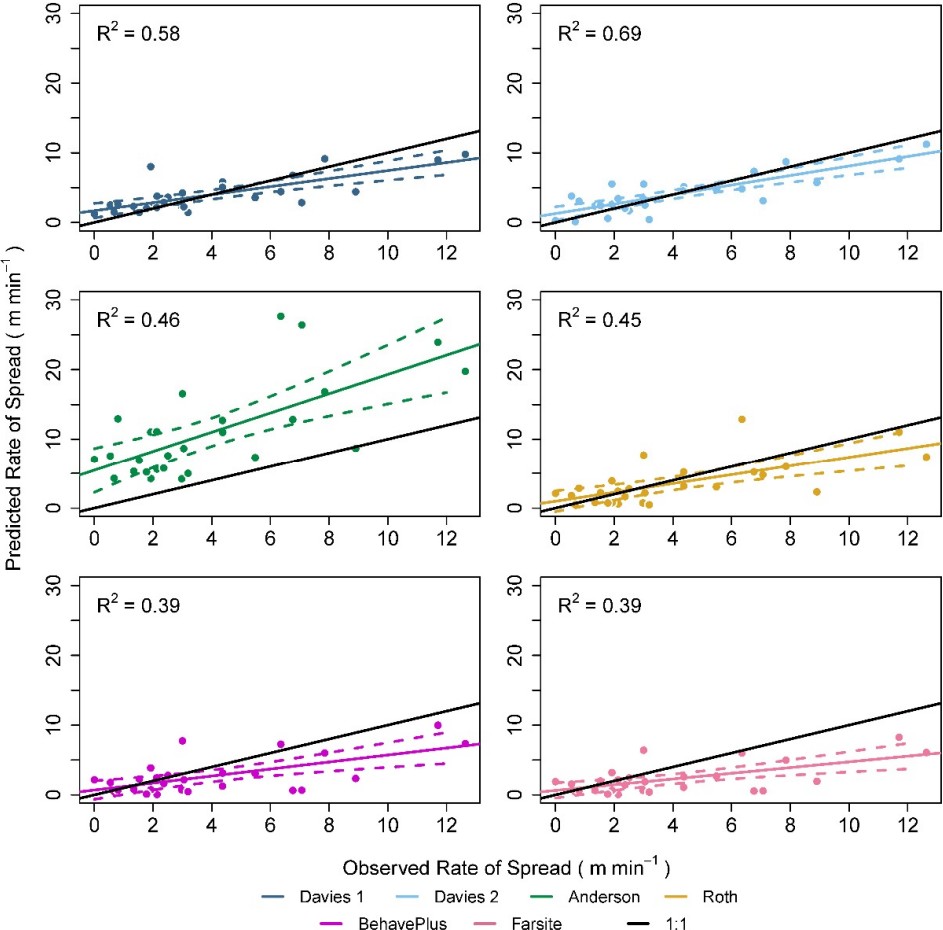

**Figure 1.** Results of linear regression analyses examining the relationship between predicted and observed rate of spread from six fire behaviour models. Coloured lines represent the regression and dashes represent 95% confidence intervals. The black line is the 1:1 LPA and individual observations of fire behaviour are shown as dots. Davies 1 and Davies 2 were built using the same fire behaviour observations; therefore, this analysis does not provide independent verification of their performance—these results are given to contextualize the performance of the other models. Further, the correction for Anderson is not shown as error was greatly increased.

For predictions of observed fires, ANOVA revealed significant effects of the fire behaviour model used and fuel class modeled, but no significant interaction between the two (Table 3). Post-hoc pair-wise comparisons confirmed that predictions from the Anderson model were significantly different to the others (Table S3 in Supplementary Material). There were no significant differences in mean predicted rate of spread between the other fire behaviour models. Roth, BehavePlus, and Farsite, all implementations of the Rothermel model, were the most similar to each other. For the fuel class factor, mean rate of spread was significantly greater in the High class, but there was no significant difference between the Medium and Low classes. From our Levene's test based on the interaction of fire behaviour model and fuel class, we found that population variances were unequal, suggesting that different prediction methods produced an unequal spread of fire behaviour within different fuel classes.

**Table 3.** ANOVA comparing the effects of the fire behaviour model (Model), fuel class (Low, Medium, or High load), and their interaction on rate of spread predictions using observed fuel structures. 'df' is n−1 degrees of freedom. 'F Value' is the test statistic computed. 'Pr (>F)' is a significance value for each test set against $\alpha$ = 0.05. N = 27.

| Response: Rate of Spread | df | F Value | Pr (>F) |
|---|---|---|---|
| Model | 5 | 28.608 | <0.001 |
| Fuel | 2 | 33.747 | <0.001 |
| Model × Fuel | 10 | 0.903 | 0.532 |

*3.2. Variation in Rate of Spread Model Sensitivity*

When examining model behaviour for the High, Medium, and Low class fuel beds across the wider range of fuel moistures and wind speeds we defined, there were significant differences within both fire behaviour model and fuel class effects. The interaction between fire behaviour model and fuel class was significant, though its effect size was much lower (Table 4). Post-hoc pair-wise comparisons indicated that there were significant differences between Anderson and all other fire behaviour models (Table S4 in Supplementary Material). There were no significant differences found between the other models, the most similar of which were Roth and BehavePlus. Significantly higher rates of spread were reported for the High fuel class but the Low and Medium classes did not differ.

**Table 4.** ANOVA comparing the effects of fire behaviour model (Model), fuel class (Low, Medium, or High load), and their interaction on rate of spread predictions for the broader range of fire weather conditions. 'df' is n − 1 degrees of freedom. 'F Value' is the test statistic computed. 'Pr (>F)' is a significance value for each test set against $\alpha$ = 0.05. N = 50.

| Response: Rate of Spread | df | F Value | Pr (>F) |
|---|---|---|---|
| Model | 5 | 230.793 | <0.001 |
| Fuel | 2 | 134.830 | <0.001 |
| Model × Fuel | 10 | 1.918 | 0.040 |

Considering the effects of live woody fuel moisture content, wind speed, and fuel class, the model sensitivities for Roth, Behave, and Farsite were very similar (Figure 2). The Davies 2 model displayed the greatest sensitivity to FMC, with reduced rates of spread as live FMC increased, but none of the Rothermel model implementations showed any meaningful change in their predictions. For all models, the high fuel class displayed a much greater response to increasing wind speed compared to the low and medium classes. The Anderson model was the most sensitive to increasing wind speed, particularly in the low fuel class where changes were more muted for the Davies models and the Rothermel



implementations. All fire behaviour models, other than Davies 2, were consistently more sensitive to changes in wind speed than changes in live woody FMC. For Davies 2, the model was more sensitive to changing live FMC when examining the Low fuel class (Figure 2). The three implementations of the Rothermel model showed very similar responses to changing fuel structure, FMC, and wind speed. The R implementation (Roth) did however display numerically different results, producing increased predictions at higher wind speeds.

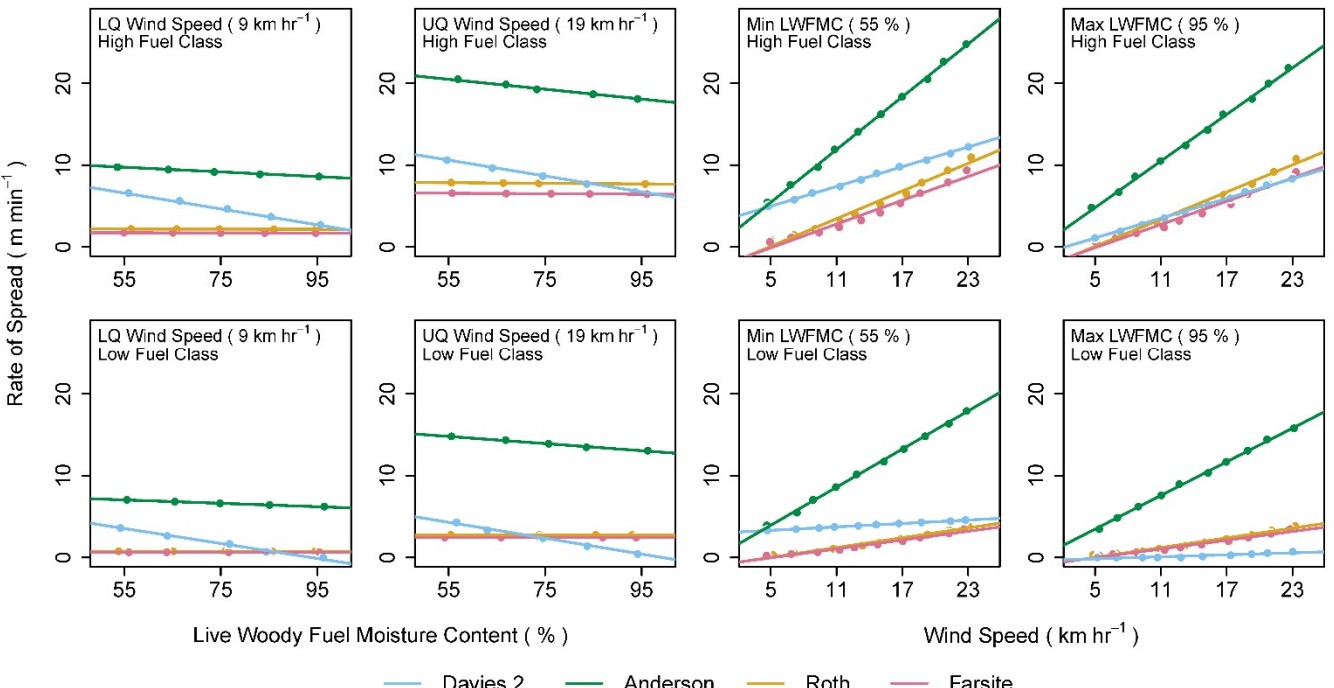

**Figure 2.** Sensitivity responses to changing fuel structural and fire weather inputs. High and Low fuel classes are aggregate fuel classes from Legg et al. [61]. Displayed fire behaviour models were selected to minimize clutter and overlaps. Full results are presented in the Supplementary Material (Table S5). Here, to gauge the interaction of fire weather parameters, lower (LQ) and upper quartiles (UQ) of wind speed were selected as constants to compare the effects of changing live woody (LW) fuel moisture contents (FMC) on the predicted rates of spread. Because a smaller effect on the rate of spread by live woody FMC was recorded, minimum and maximum values were selected for live woody FMC to compare the effects of changing wind speeds on the predicted rates of spread. A jitter was applied on the *x*-axis for clarity.

## 4. Discussion

The experimental fires used to evaluate fire rate of spread models capture a broad array of fuel conditions from multiple stages of the *Calluna* growth cycle and a range of fire weather conditions typical during the legal prescribed burning period in the UK [45]. This data could not be used to make an independent evaluation of the empirical models we examined, but the assessment here provides useful context against which to consider the performance of the Rothermel model [35,44]. Extrapolating purely empirical models is, however, fraught with risks and, in our case, the available experimental data do not capture, for example, the full diversity in *Calluna* fuel types found in northern heathlands, nor the variation in fire weather conditions seen throughout the year. For example, previous research has demonstrated that during extended periods of hot and dry weather, layers of moss and litter below the *Calluna* canopy may be ignited, significantly altering burn severity and, potentially, fire behaviour [63,74]. Other work has shown that live fuel moisture contents can drop to as low as 41–44% during conditions when the ground is frozen in winter and early spring [32,75]. Conditions such as these are associated with

significantly increased wildfire risk but are not captured in existing empirical models of fire behaviour.

Given the wide variety of fuel and fire weather conditions encountered in northern heathlands, and the growing challenge of managing wild and prescribed fires in these ecosystems, it is imperative that managers have access to robust and broadly applicable fire hazard and fire behaviour prediction tools [24]. The basis of the Rothermel model in fundamental physical processes [51], its widespread adoption in multiple fuel types, and its implementation in multiple manager-orientated tools makes it a potentially important solution to this problem [55,58,76]. Our results demonstrated that the Rothermel model in R, BehavePlus, and Farsite produced rates of spread that were broadly representative of observations, demonstrating their relative suitability for use in oceanic heathlands. RMSEs for all fuel structures were relatively similar within each fire behaviour model. $R^2$ values for the Rothermel implementations (range = 0.39 to 0.46) indicated a somewhat noisy fit to observed rates of spread and thus a considerable degree of uncertainty when using this model. In comparison, the fully empirical Anderson model had a comparatively high $R^2$ but showed substantial and consistent over-prediction, especially across a broadened range of possible fire weather, possibly due to its generation using longer fire fronts and its more extensive use of shrublands in drier and warmer ecosystems. This further highlights the importance for empirical models to be calibrated to specific ecosystems and potential climatic and weather conditions. We can summarize the predictive ability of the generalizable fire models as follows:

1. Rothermel implementations have moderate precision, but lower bias compared to Anderson. Bias increases for higher rates of spread where there appears to be consistent under-prediction of observed fire behaviour.
2. Anderson is comparatively precise but biased, producing rates of spread that are dramatically higher than those observed in fires with ignition line lengths used in managed burning.
3. With correction of our observed rates of spread for differences in ignition line length, overprediction by Anderson was reduced at the cost of considerably greater RMSE and a lower $R^2$ correlation value.

Broadly speaking, the models showed similar sensitivities to changes in wind speed, though this was somewhat dependent on the fuel class considered. The Rothermel implementations differed from the empirical models with regards to the effect of live FMC (where included as a predictor)—Rothermel implementations essentially showed no effect. Other studies have noted the Rothermel model's lack of sensitivity to live fuel moisture contents [77], especially with respect to canopy foliage [78], and that empirical models are more effective at representing the moisture dampening effects of live fuels [79,80]. This is something of a concern as it may lead to underestimates during high-risk conditions seen in winter and spring that are driven by dramatic crashes in live FMC [32,75]. Furthermore, the Rothermel model does not consider convective processes governing fire spread [81], an issue that could be addressed with physics-based models [51] and which may explain much of the noise in comparisons with observed fire behaviour. Nevertheless, other than for Anderson fire, behaviour predictions between models were not significantly different. This indicates that Roth, BehavePlus, and Farsite produce statistically similar fire behaviour, using real-life fuel structures, both to each other and to empirical models calibrated for oceanic heathlands (Davies 1 and 2).

## 5. Conclusions

Our results indicate that the Rothermel model could be cautiously utilized as a tool for future fire research and land management objectives. Although Rothermel's predictions align tolerably well with empirical models generated for oceanic heathlands [21], issues of under and overprediction must be considered when determining the right course of action in any project or management [82]. We thus encourage further research to validate and calibrate the model over a wider range of heath, moorland, and peatland fuel types, as

well as across more diverse fire weather scenarios. Changes in live fuel moisture content can significantly alter fire behaviour [83], such as those present in *Calluna*-dominated heathlands [61]; however, the Rothermel model shows limited responsivity to this driver. If fire behaviour is to be forecasted, further research must seek to develop models for live fuel moisture and better constrain its effects on fire spread. Our study was purposefully conducted with flat terrain and for steady-state headfires to simplify comparisons. Future studies should investigate the impacts of varying geography and for different sections of a fire (e.g., backfires and flanks), since fires on slopes and backing fires have been observed to display different fire behaviours [84]. Finally, collecting observations of fire behaviour from case-study wildfires and experimental fires with longer ignition line lengths is important. Such data will help to evaluate these models in operational settings and develop more robust relationships between fire behaviour predictions and direct observations.

**Supplementary Materials:** The following supporting information can be downloaded at: https://www.mdpi.com/article/10.3390/fire5020046/s1, Figure S1: Results of linear regression analyses examining the relationship of Anderson v. observed rates of spread; Table S1: Fuel structural characteristics for all prescribed burns; Table S2: Fire behaviour summary table used to generate model performance testing versus observed fire behaviour; Table S3: TukeyHSD post-hoc analyses following the ANOVA for rate of spread versus fire behaviour model and fuel class for observed fire behaviour comparisons; Table S4. TukeyHSD post-hoc analyses following the sensitivity analysis ANOVA; Table S5. Fire behaviour summary table used to generate all sensitivity analyses. All data and analytical scripts used in the paper are also provided in the Supplementary Material.

**Author Contributions:** Conceptualization, C.D.M.-D. and G.M.D.; methodology, C.D.M.-D. and G.M.D.; formal analysis, C.D.M.-D.; data curation, G.M.D.; writing—original draft preparation, C.D.M.-D.; writing—review and editing, C.D.M.-D. and G.M.D.; supervision, G.M.D.; funding acquisition, C.D.M.-D. and G.M.D. All authors have read and agreed to the published version of the manuscript.

**Funding:** This research was funded by the Scottish Government and Scottish Natural Heritage through the FireBeaters project, the Game and Wildlife Conservation Trust, the Natural Environment Research Council (NER/S/C/2001/06470), and the Research Council of Norway (Grant number 298993).

**Institutional Review Board Statement:** Not applicable.

**Informed Consent Statement:** Not applicable.

**Data Availability Statement:** The raw data, manipulated data, and R scripts used in this study's analyses are openly available in FigShare at (https://doi.org/10.6084/m9.figshare.16635919) (accessed on 20 January 2022).

**Acknowledgments:** A large number of individuals contributed to fieldwork that allowed collection of the fuel bed and fire behaviour data used in this study. In particular, we thank Colin Legg, Adam Smith, Scott Newey, David Howarth, Alan Kirby, Harry Robertson, Carol Smithard, Isla Graham, Ellie Watts, Ella Steele, Bill Higham, Teresa Valor Ivars, and Elaine Boyd. We especially thank the FireBeaters; the Scottish Fire Danger Rating System; and the dynamic research teams at the University of Edinburgh, James Hutton Institute, the University of Bergen, and Western Norway University of Applied Sciences for their collaboration and assistance, namely Colin Legg, Michael Bruce, Angus MacDonald, Rory Hadden, Guillermo Rein, Jason Owen, Andy Taylor, Torgrim Log, Liv Guri Velle, Vigdis Vandvik, Anna Marie Gjedrem, and many others. We are extremely grateful to Ralia Enterprises and Whitborough Estate for allowing us to use their land for the experimental burning trials and to their staff and gamekeepers for their generous logistical support and advice. Finally, we would like to thank Stephen Matthews and Roger Williams for their detailed input during the development and review stages of this research. We thank the anonymous reviewers whose comments helped improve the paper.

**Conflicts of Interest:** The authors declare no conflict of interest.

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
