# Peer review of "Evaluating the Performance of Fire Rate of Spread Models in Northern-European Calluna vulgaris Heathlands"

_fire, doi:10.3390/fire5020046_

Round 1
Reviewer 1 Report
Review on Ms. Ref. No.: fire-1661012
Evaluating the performance of fire rate of spread models in northern-European Calluna vulgaris heathlands
Charles D. Minsavage-Davis and G. Matthew Davies
Recommendation: Accept after minor revision
- Summary
The authors of the submitted manuscript evaluate the outputs of several empirical and quasi-empirical prediction models for fire rate of spread models in northern-European Calluna vulgaris heathlands, as well as their varying software implementations, against observations of fire behavior. Good experimental fire methodology with 27 fuel structures was used compared with measured rates of spread with predictions from three different implementations of the Rothermel model: the ‘ros’ function of the R ‘Rothermel’ package, BehavePlus, and Farsite. Good work!
- Major issues
No major issues.
- Minor issues
- The literature review part in the Introduction should be improved. For example, it is a pity for your paper, not to step on the wildland fire research of ADAI_CEIF Forest Fire Research Center in Portugal (adai.pt). Including the full-scale wildfire test they perform.
- I recommend the authors form a Conclusion section. In that way, the paper looks better and finished. Why not put the latest paragraph from the Discussion to a Conclusion?
- Opinion
I have read the first-round review comments and their answers with an interest. However there are some comments from the first reviewer which make sense (and are revised accordingly by the authors), generally, it looks like a negative criticism rather than constructive feedback which is the purpose of peer-review. In this respect, I like and stand behind the authors’ in-depth answers and appreciate their work.
Despite the remarks above, I assess the overall manuscript as very good and I recommend it for acceptance after minor revision.
Kind regards!
Author Response
Authors note - reviewers comments are in red, our response is in black below
The literature review part in the Introduction should be improved. For example, it is a pity for your paper, not to step on the wildland fire research of ADAI_CEIF Forest Fire Research Center in Portugal (adai.pt). Including the full-scale wildfire test they perform.
This study is mainly focused on a dominant landcover in British Uplands and coastal Norway (heathland shrub ecosystems), which also coverage in northern Europe. For this reason, we wish to highlight the major issues with predicting fire behavior in these northern areas with cool, wet climates while addressing southern Europe in the context of global fire. We have done this in paragraph 1 of the intro, which we have revised to include Portugal in a broader sense. This then funnels into the UK and Norway specifically in paragraph 2. Further, our discussion now includes a more comprehensive relationship with work done by other groups in quantifying characteristics of fires in Europe. We have added several citations to work completed by researchers associated with the ADAI_CEIF Forest Fire Research Center
I recommend the authors form a Conclusion section. In that way, the paper looks better and finished. Why not put the latest paragraph from the Discussion to a Conclusion?
We have edited the end of the paper to include a conclusion section formed from the last paragraph of our previous discussion, with additional insights that may help future readers to understand our recommendations as well as guidance for further research.
Reviewer 2 Report
This is a comparative study that considers different empirical models for fire spread when applied to northern-Europe Calluna vulgaris heathlands. The empirical models are two models by Davies, one model by Anderson, an R implementation of Rothermel's spread model, Beehave, and Farsite. Each of these models can be considered as a different implementation of the classical Rothermel spread model.
This is a nice paper that is appropriate for Fire. A few comments below should be addressed before it is published
- In section 2.1, several speeds are given in different units (km/h and m/min). Can a single unit be used so that it is easy for readers to quickly compare the numbers?
- I understand that equations (1), (2), and (3), are all different ROS equations, but because there is no equal sign in each of these equations, it could lead to confusion. Maybe something as easy as ROS_{D1} and ROS_{D2} for equations (1) and (2) would help.
- The formatting of the text immediately after equations (1)-(5) can be improved. I recommend not centering the text with the larger margins, but instead simply continue the text as "where $h$ is the mean Calluna height (m), ...
- Equation (4) should be put closer to where it is referenced in the text.
- Not enough information about the Corrected model in Table 2 is provided.
Typos:
- due to a projected increase *of* severe fire weather.
- were developed with *the* same experimental fire data we used
- There is a mixture of present and past tense throughout. I would recommend using only the present tense. To give an example, the paragraph before equation (5) might sound better in the present tense.
Author Response
Authors note: reviewers comments are in red, our responses are below
In section 2.1, several speeds are given in different units (km/h and m/min). Can a single unit be used so that it is easy for readers to quickly compare the numbers?
We have ensured that the paper only uses km hr-1 for wind speed and m min-1 for fire rate of spread. The main reason for this difference is that wind speeds are generally much higher than fire rates of spread, and these values are far too different to represent effectively using one unit. Further, within wildland fire management professions, the standard for reporting fire rate of spread tends to be m min-1 and for wind speed is either m s-1 or km hr-1.
I understand that equations (1), (2), and (3), are all different ROS equations, but because there is no equal sign in each of these equations, it could lead to confusion. Maybe something as easy as ROS_{D1} and ROS_{D2} for equations (1) and (2) would help.
The formatting of the text immediately after equations (1)-(5) can be improved. I recommend not centering the text with the larger margins, but instead simply continue the text as "where $h$ is the mean Calluna height (m), ...
Equation (4) should be put closer to where it is referenced in the text.
The three comments above all pertain to the formatting of presented equations. We have reformatted our equations to represent the changes suggested in this reviewer’s comments.
Not enough information about the Corrected model in Table 2 is provided.
This has been addressed at the end of the caption for Table 2.
Typos:
- due to a projected increase *of* severe fire weather.
- were developed with *the* same experimental fire data we used
We have addressed both of these minor issues
There is a mixture of present and past tense throughout. I would recommend using only the present tense. To give an example, the paragraph before equation (5) might sound better in the present tense.
This seems to be an issue of preference and we believe that our current grammatical style does not affect readability or interpretation of our study in any way. Much of our intro is in present tense to say "this is the current state of fire behaviour prediction in our target ecosystem" and almost all of the methods are in past tense, which we believe strikes an appropriate distinction for reporting how we approached our analyses.
Reviewer 3 Report
Manuscript Number: fire-1661012
Full Title: Evaluating the performance of fire rate of spread models in northern-European Calluna vulgaris heathlands
This research evaluates the outputs of several empirical and quasi-empirical prediction models, as well as their varying software implementations, against observations of fire behaviour. The paper is interesting and I have several concerns that in my view should be addressed prior to consideration for publication in fire. The comments are listed below:
- What’s the main characteristic of the burning in northern-European Calluna vulgaris heathlands?
- More description, i.e., advantage, disadvantage, limitations, etc., should be given for the Rothermel model, generic shrubland empirical model and other listed models.
- The R2 number is relatively low in Fig. 1, please give the reason and make sufficient description.
- The figures are too small in Fig. 2.
- Why the generic shrubland empirical model overpredicts observed rates of spread for prescribed burns?
- Please give recommendations about the listed empirical models and software tools for evaluating potential fire behaviour and assess risk in heathland and moorland landscapes.
Author Response
Authors note: reviewers comments are shown in red, our responses are provided below
What’s the main characteristic of the burning in northern-European Calluna vulgaris heathlands?
We believe this comment regards the use of managed burning in northwestern Europe. This has been addressed extensively in paragraph 2 of the intro with specific reference to the current changes in policy, impacts of previous wild and managed fire in the region and potential future issues in fire risk. Furthermore, we give more description of what work has been done to characterize fuels and fire in these ecosystems in paragraph 3 with a pointer toward a lack of generalizable fire behaviour prediction modeling (expanded upon in paragraph 4). In general, we have also reworked some of the clunkier parts of the intro and some more confusing regional associations of our dominant landcover type. We have added a brief summary of the characteristics of managed burning and fire behaviour (lines 55-60)
More description, i.e., advantage, disadvantage, limitations, etc., should be given for the Rothermel model, generic shrubland empirical model and other listed models.
An evaluation of where the current empirical models lack, and how the Rothermel model seeks to fill those gaps, has already been given in paragraphs 3 and 4 of the intro. We have also included a more explicit comparison of the models we used in our analyses in lines 97-100. This should easily translate to the description of model terms in our methods section in lines 164-202.
The R2 number is relatively low in Fig. 1, please give the reason and make sufficient description.
We have addressed this in our discussion on lines 402-412.
The figures are too small in Fig. 2.
We have addressed this issue.
Why the generic shrubland empirical model overpredicts observed rates of spread for prescribed burns?
This has been addressed in lines 384-389.
Please give recommendations about the listed empirical models and software tools for evaluating potential fire behaviour and assess risk in heathland and moorland landscapes.
This has been addressed in our conclusion, specifically at the beginning.
Round 2
Reviewer 3 Report
The authors have replied my comments properly and I suggest acceptance of this manuscript.
This manuscript is a resubmission of an earlier submission. The following is a list of the peer review reports and author responses from that submission.
Round 1
Reviewer 1 Report
- I have two major concerns about the methodology: a) 20X30 or 30X30 m plots are not sufficient for any fire to achieve 'steady state' conditions, moreso in prescribed burns with low windspeeds, and b) your fuelbed consists almost completely of live vegetation with no pronounced litter floor. Both Rothermel's (1972) and Anderson's models are specifically based, developed and validated in dead, dry and fine fuelbeds.
- The study relies heavily on Legg et al. 2007 [57] practically unpublished data.
- It is well established in the fire scientific community that Rothermel's model provides the most reliable fire behavior predictions since the 80's. As a mater of fact, FARSITE and FLAMAP are based on Rothermel's model. This paper provides no additional insight.
- There are no information regarding the other fire parameters such as flame length, flame height and fireline intensity.
Reviewer 2 Report
Review all the work with a practical application case with os simulation software.